# Experienced cognitive load in the emergency department. A prospective study

**Yael Appelboom**[1], **Yvonne Groenen**[2], **Dirk Notten**[3], **Anique De Bruin**[4], **Jacqueline Buijs**[1], **Harm R. Haak**[5], **Hella F. Broggreve**[6], **Lars Lambriks**[6], **Patricia M. Stassen**[6]*

1 Department of Internal Medicine, Zuyderland Medical Centre, Heerlen, Netherlands, 2 Emergency Department, St. Jans Gasthuis, Weert, Netherlands, 3 Intensive care, Zuyderland MC, Heerlen, Netherlands, 4 Department of Educational Development and Research, School of Health Professions Education, Maastricht University, Maastricht, The Netherlands, 5 Department of Internal Medicine, Máxima Medical Centre, Maastricht University, CAPHRI School of Public Health and Primary Care, Ageing and Long-Term Care, Maastricht University, Maastricht, The Netherlands, 6 Department of Internal Medicine, Maastricht University Medical Centre+, CARIM School for Cardiovascular Diseases, Maastricht University, Maastricht, The Netherlands

* p.stassen@mumc.nl

## Abstract

### Background and importance

The emergency department (ED) is a hectic place, where many critically ill patients are treated. For residents working in the ED, this environment may be demanding.

### Objectives

The aim of this study was to investigate the [1] cognitive load experienced by residents working in the ED, and [2] differences in cognitive load during the day.

### Methods

In this multicentre, prospective study in three EDs in the Netherlands, the experienced cognitive load was graded by residents on three scales, agreed upon during Delphi meetings: the complexity (low-high: 1–3), mental effort (low-high: 1–9) and comfortability scale (low-high: 0–100%). We applied the scores per decision, 1-hour and 2-hour intervals, patient and shift.

### Main results

We observed 14 residents and analysed 74 1-hour and 45 2-hour intervals, 79 patients, 24 shifts and 592 separate decisions. The experienced cognitive load per decision was low. In contrast, the cognitive load was higher per 2-hour interval (mental effort: median 4.0 (IQR 4.0) and comfortability 80% (IQR 20)) and per shift (mental effort: median 5.5 (IQR 4.0) and comfortability 80% (IQR 20). Complexity was low for all measurements. Mental effort rose from 17h onwards higher values, whereas a decrease in comfortability was seen from 21h onwards.

**Funding:** The author(s) received no specific funding for this work.

**Competing interests:** The authors have declared that no competing interests exist.

## Conclusion

From 17h onwards, residents working in the ED experienced rather high mental effort and reported feeling not optimally comfortable when making decisions. The mental effort was highest between 21-23h. This was found when cognitive load was measured per 2-hour interval and per shift, but not per decision. This study may provide an insights to optimise cognitive load by reorganisation of the ED.

## Introduction

The emergency department (ED) is an often hectic and crowded department where many critically ill patients are treated. Doctors have to make many decisions, often based on incomplete or complex information. Moreover, doctors are repeatedly interrupted in their thought process [1, 2].This combination of factors, in combination with diagnostic uncertainty, may lead to high experienced cognitive load and consequently, to a decrease in work performance [3, 4].

Cognitive load is what trainees experience when they engage in the process of learning and storage of information in the human brain [5]. It can be divided into three types: the intrinsic, the extraneous and the germane cognitive load. The intrinsic cognitive load is determined by the complexity of a task and the person's knowledge. The extraneous cognitive load is created by the way a task is presented, and the germane load is the result of the construction and automation of schemes, e.g. patterns that are recognized. All three types of cognitive load play a considerable role for those working in the ED.

The available studies in the ED focus on workload rather than on cognitive load. These studies show that the doctor's level of experience, the total number of patients during a shift and the overall complexity of the cases influence the workload [6]. In addition, there is often a peak of patient influx around the changing of shifts, leading to fluctuations in workload [7, 8].

As an excess in experienced cognitive load leads to decreases in work performance [4] and learning efficiency [9], it may be insightful to objectify the overall and the variation in cognitive load, as experienced by residents working in the ED. Knowing when the experienced cognitive load is high may help to organise the ED and, more specifically, the shifts in the ED.

In this prospective, multicentre study, we aimed to investigate the experienced cognitive load of residents working in the ED and its variation over the day. The cognitive load was measured by assessing the complexity, mental effort and comfortability. Our hypothesis was that residents experience a high cognitive load when working at the ED.

## Methods

### Setting and study design

This study was a multicentre, prospective, observational study, performed at three EDs in the Netherlands. The study was conducted in Maastricht University Medical Centre+ (MUMC+), Zuyderland Medical Centre Sittard (Zuyderland) and Maxima Medical Centre Veldhoven (MMC). MUMC+ is a university hospital, whereas Zuyderland and MMC are large teaching hospitals. The three EDs are comparable in terms of procedures and workload. The residents, rather independently, see all kinds of patients with internal medicine problems. After their assessment and interpretation of all ancillary investigations they call their supervisor, which is present during office hours. During out-of-hours, the contact is mostly by phone. In MUMC,

the supervisor is present until 8 pm, while in the other two hospitals, emergency physicians assist the residents if necessary.

Internal medicine residents graded the experienced cognitive load during their shifts, using three different scales. Grades were subsequently recorded by two independent investigators. The STROBE checklist for reporting data was used to verify all significant reported items [10]. The Medical Ethics Committee of the MUMC+ approved the study protocol (METC 2019–1149). All residents who participated provided written informed consent.

### Study participants

The study participants were residents in internal medicine who worked multiple shifts in the three different EDs. Observations took place between September and December 2019.

### Cognitive load measurements

To date, there is no gold standard to measure experienced cognitive load in the ED. Therefore, we first conducted a pilot study, during which the experienced cognitive load was observed and graded. For further adjustments of the study design, three Delphi meetings were held, with medical experts from the participating centres: four attending physicians in acute internal medicine, one fellow in acute internal medicine, a psychologist, a health science researcher and two investigators [11]. The Delphi meetings were used to determine the study design, agree upon the scales used to investigate the experienced cognitive load and analyse the preliminary results.

During the Delphi meetings, 3 scales were chosen to measure cognitive load in the ED. As measuring cognitive load in the ED was not done before, we chose more than one scale. The first scale was the complexity scale (score 1–3), where 1 represents a simple and 3 a highly complicated problem [8]. The second scale was the mental effort scale (score 1–9), where 1 is scored for the lowest, and 9 for the highest mental effort [12]. This scale is widely used. The third scale was the comfortability scale, which is a percentage scale, with 100% being completely comfortable and 0% being not at all comfortable [13].

Five types of measurements for experienced cognitive load were conducted. During the first 7 weeks, the experienced cognitive load was measured per decision in one centre (MUMC+) ("Decision measurements"). The subsequent 6 weeks, the cognitive load was measured per hour, 2 hours, patient and per shift in the three centres ("Complementary measurements").

### Data collection

The collected data were analysed anonymously. Two investigators independently collected data from the same shifts for three consecutive days, to make sure they recorded the data in the same way. The residents were asked regarding the cognitive load by the investigators immediately after the decision was made, or immediately after the different time-intervals, the timing depending on the type of measurement. For the decision measurements, they collected data between September 2019 and November 2019, and for the complementary measurements, between November and December 2019. The decisions observed during this period were not included in the decision measurements. The median number of decisions during the day, however, was calculated using all observed decisions during both the decision- and complementary measurements.

A weekly meeting of the investigators and members of the study team was held to make sure the collected data were aligned. Day shifts were set from 9h to 16h, afternoon shifts from 13h to 20h, and evening shifts from 13h to 23h.

The following items were retrieved: resident characteristics (years of experience), patient characteristics (sex and age), and decision characteristics (time, type of decision, subject of the decision). In addition, we recorded the following cognitive load scores: complexity, mental effort and comfortability score per decision, time interval (1-hour and 2-hour), patient and complete shift.

## Data analysis

IBM SPSS Statistics 25 (IBM Corp., Armonk, USA) was used. Descriptive statistics were used to calculate means and standard deviations (SD) and medians with interquartile ranges (IQR), depending on the distribution of the data. Medians were calculated for the cognitive load scales with regard to both the decision and the complementary measurements. The number of decisions per hour was corrected by dividing the number of decisions through the number of times that the particular hour was observed.

To test for differences in experienced cognitive load between the five types of measurements, the Mann Whitney U and the Kruskal Wallis test were applied. In addition, the 2-hour measurement was used as reference for the distribution of scores during the day and was compared with the four other types of measurements (decision, 1-hour, patient, shift). The correlation between the three cognitive load scales was calculated using the Spearman correlation coefficient. P-values $<0.05$ were considered statistically significant.

## Results

In total, 45 shifts, 14 residents and decisions concerning 207 patients were observed (Table 1). The observed residents had a median of 1.5 (IQR 4.0) years of experience. Of the 207 patients, the median age was 68 years (IQR 26.5) and 109 (52.7%) were male.

A total of 1364 decisions were observed of which 592 individual decisions were analysed during the first stage of the study (decision measurements, Table 2). Of the 45 observed shifts, 24 were assessed during the complementary measurements. During these shifts, 74 1-hour, 45 2-hour intervals and 79 patients were analysed.

A median number of decisions of 27 (IQR 21) per day were recorded when taking into account all 1364 observed decisions. Between 10h and 12h, there was a first peak in the number of decisions, followed by a decrease around midday (Fig 1). Most decisions were made between 17 and 18h.

### Decision measurements

Of the 1364 decisions, 592 individual decisions were further analysed during 22 day and 2 evening shifts in the MUMC+. Most decisions were categorized as either "Requesting

**Table 1. Characteristics of residents and shifts.**

|  | Centres | | | |
|---|---|---|---|---|
|  | MUMC+ | MMC | Zuyderland | Total |
| **Residents (n)** | 8 | 3 | 3 | 14 |
| **Experience (median, years)** | 3.5 (IQR 6.5) | <1* | 1* | 1.5 (IQR 4) |
| **Shifts (n, (%))** | 35 | 5 | 5 | 45 |
| **Day** | 27 (77.1%) | - | 5 (100%) | 27 (60.0%) |
| **Afternoon** | - | 4 (80.0%) | - | 9 (20.0%) |
| **Evening** | 8 (22.9%) | 1 (20.0%) | - | 9 (20.0%) |

*No IQR range could be calculated due to low number of residents observed

IQR = interquartile range

**Table 2. Number of observations per centre and per measurement method.**

| n (%) | Centres | | | | | |
|---|---|---|---|---|---|---|
| | MUMC+ | | MMC | | Zuyderland | Total |
| | Day | Evening | Afternoon | Evening | Day | |
| Decision Measurements | | | | | | |
| | 505 (85.3%) | 87 (14.7%) | - | - | - | 592 |
| Complementary Measurements | | | | | | |
| 1-hour | 32 (43.2%) | 14 (32.4%) | 13 (17.6%) | - | 11 (14.9%) | 74 |
| 2-hour | 10 (22.2%) | 32 (31.1%) | 8 (17.8%) | 4 (8.9%) | 9 (20.0%) | 45 |
| Patient | 25 (31.6%) | 32 (40.5%) | 17 (21.5%) | 5 (6.3%) | - | 79 |
| Shift | 7 (29.2%) | 7 (29.2%) | 4 (16.7%) | 1 (4.2%) | 7 (29.2%) | 24 |

additional investigations" (n = 146, 24.7%), "Prescribing medication" (n = 101, 17.1%) and "Evaluation of results" (n = 93, 15.7%) (Table 3).

The median scores for the cognitive load scales were 1.0 (IQR 0) for complexity (1–3), 1.0 (IQR 2.0) for mental effort (1–9) and 100% (IQR 0) for comfortability (0–100%) (Table 4). Both the complexity (n = 519, 87.7%) and the mental effort (n = 310, 52.4%) scales were most often scored as 1. With respect to comfortability, 100% (76.0%, n = 450) was most often scored.

## Complementary measurements

**Time interval measurements (1-hour and 2-hour).** For the 1-hour measurements, the median score was 1.0 (IQR 1) for complexity, 3.0 (IQR 5.0) for mental effort and 90% (IQR 20)

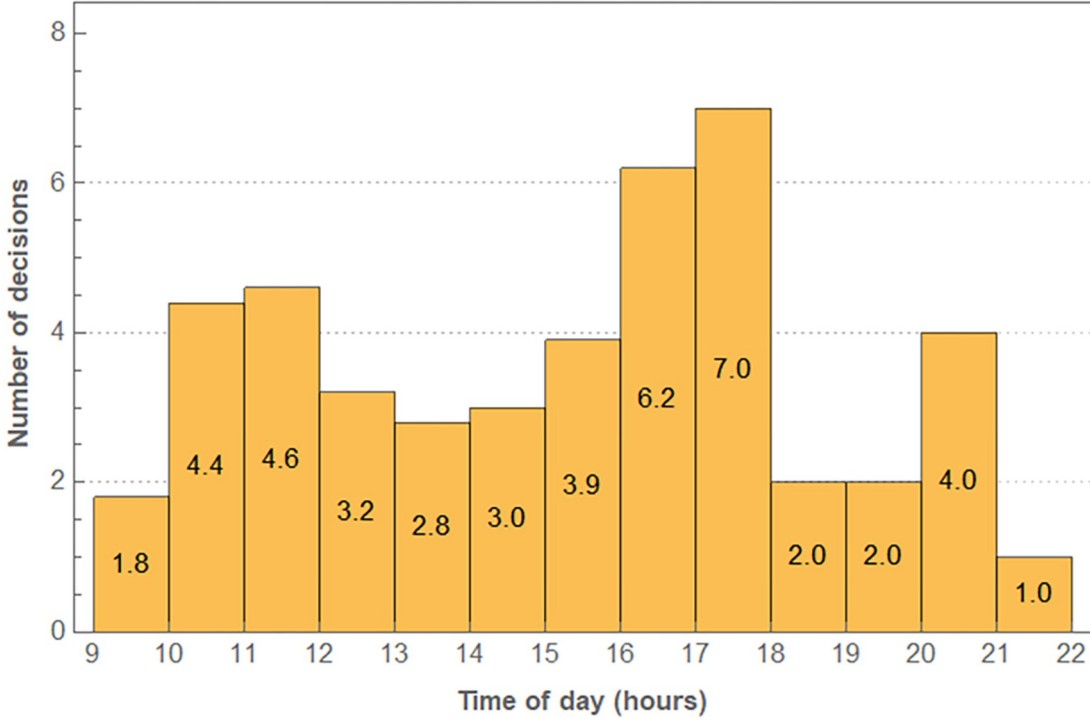

**Fig 1. Median number of decisions per hour.** The median number was corrected for the times a specific hour was observed.

**Table 3. Types of decisions.**

| Type of decisions | Frequency (%) |
|---|---|
| Patient registration | 9 (1.5) |
| Delegation of tasks to others | 87 (14.7) |
| Physical examination and interpretation | 60 (10.1) |
| Requesting additional investigations | 146 (24.7) |
| Evaluation of results | 93 (15.7) |
| Diagnosis | 3 (0.5) |
| Prescribing medication | 101 (17.1) |
| Giving instructions to colleagues | 9 (1.5) |
| Asking a consultation of another specialism | 10 (1.7) |
| Asking for an early evaluation by supervisor | 10 (1.7) |
| Discharge from Emergency Department | 52 (8.8) |
| Miscellaneous | 12 (2.0) |

for comfortability. Although the mental effort score was higher during the evening compared to during the day, this was not significantly different (p = 0.066, Fig 2). The comfortability score, however, was significantly lower after 15h compared to before (80 vs 90%, p = 0.007).

For the 2-hour measurements, the median scores were 1.0 (IQR 1.0) for complexity, 4.0 (IQR 4.0) for mental effort and 80% (IQR 20; 70–82.5%) for comfortability. Although the mental effort between 17h-19h (7.0, IQR 6.0) and the complexity between 19h-21h (2.0, IQR 1.0) were higher compared to those between 15h-17h (4.0, IQR 4.0) and 17h-19h (1.5, IQR 1.0), these differences were not significant (mental effort p = 0.648; complexity p = 0.752, Fig 2).

## Comparison of measurements

The median complexity for all five types of measurements was comparable (1.0), although differences in the distribution of the scores resulted in significant differences (p<0.0001, Table 5).

The mental effort score was significantly higher for the 2-hour measurement (4.0) than for the decision (1.0, p<0.001), the patient (3.0, p = 0.041) and the 1-hour measurement (3.0, p = 0.009), but not for the shift measurement (5.5, p = 0.675).

**Table 4. Scores for the three cognitive load scales per decision.**

| Scales | Median (IQR) | Number |
|---|---|---|
| Complexity (1–3) | 1.0 (0) | |
| 1 | | 519 (87.7%) |
| 2 | | 58 (9.8%) |
| 3 | | 3 (2.5%) |
| Mental Effort (1–10) | 1.0 (2.0) | |
| 1 | | 310 (52.4%) |
| 2–4 | | 230 (38.9%) |
| >4 | | 52 (8.8%) |
| >6 | | 27 (4.6%) |
| Comfortability (1–100%) | 100 (0) | |
| 100% | | 450 (76.0%) |
| 90–99% | | 56 (9.5%) |
| 70–89% | | 68 (11.5%) |
| <70% | | 18 (3.0%) |

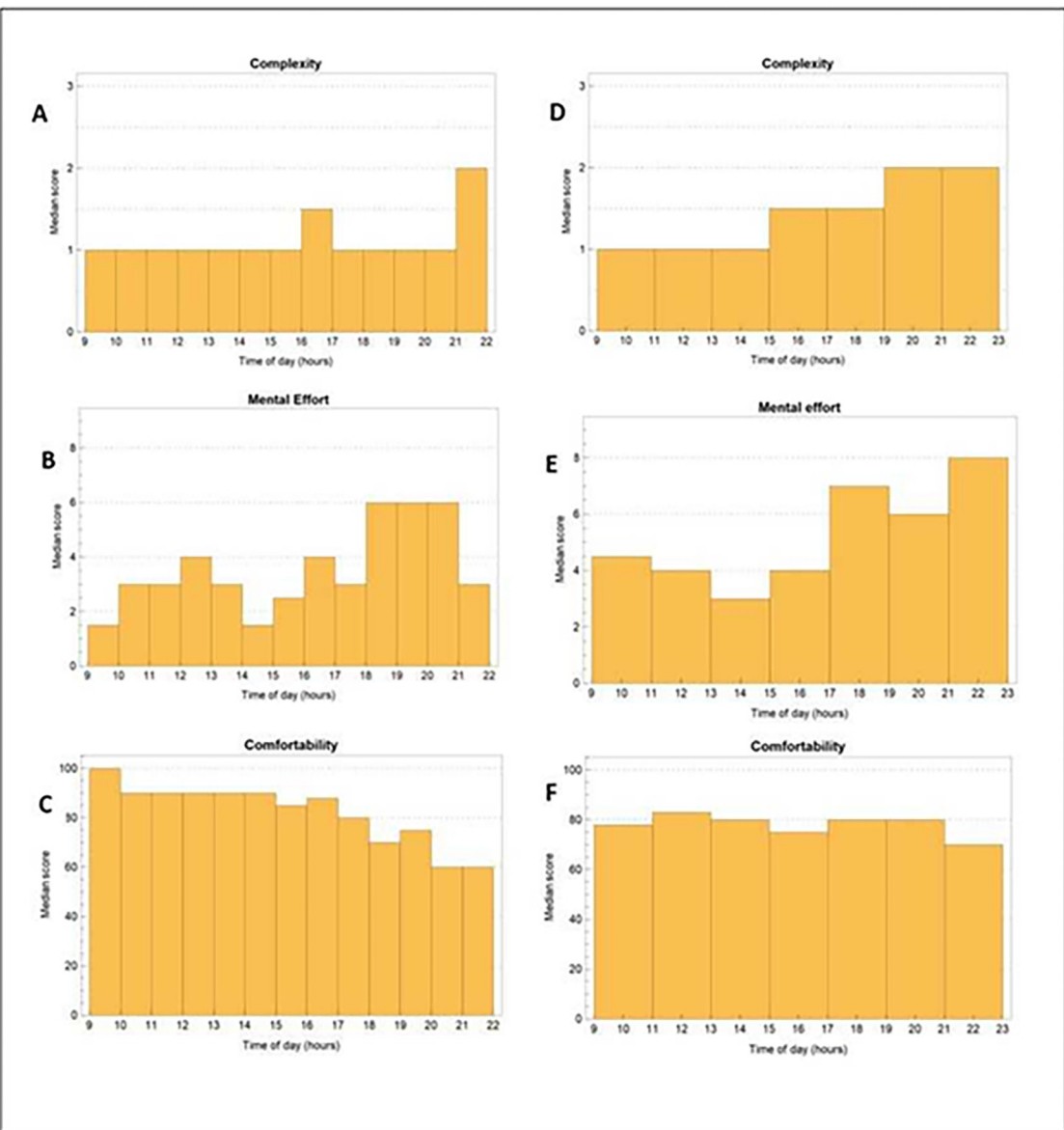

**Fig 2. Median complexity, mental effort and comfortability for the 1-hour and 2-hour measurements.** (A) median complexity,
(B) mental effort and (C) comfortability for the 1-hour measurements. (D) median complexity, (E) mental effort and (F)
comfortability for the 2-hour measurements.

The comfortability score was significantly lower for the 2-hour (80%) than for the decision
(100%, p<0.001), patient (80%, p = 0.017) and 1-hour measurement (90%, p<0.001), but not
for the shift measurement (80%, p = 0.495).

## Correlations between the three scales

There was a positive correlation between complexity and mental effort (r = 0.530, p<0.001),
and a negative correlation between both complexity and comfortability (r = -0.496, p<0.001),
and mental effort and comfortability (r = -0.656, p<0.001).

**Table 5. Median cognitive load scales for all types of measurements.**

| Measurements | Per 2-hour interval (reference) | Per decision | Per patient | Per 1-hour interval | Per shift | p value** |
|---|---|---|---|---|---|---|
| median or % (IQR) | N = 45 | N = 592 | N = 79 | N = 74 | N = 24 | |
| **Complexity (1–3)** | 1.0 (1.0) | 1.0 (0) | 1.0 (1.0) | 1.0 (1.0) | 1.0 (1.0) | <0.0001 |
| Comparison* | | p<0.001 | p = 0.451 | p = 0.054 | p = 0.831 | |
| **Mental effort (1–9)** | 4.0 (4.0) | 1.0 (2.0) | 3.0 (5.0) | 3.0 (5.0) | 5.5 (4.0) | <0.0001 |
| Comparison* | | p<0.001 | p = 0.041 | p = 0.009 | p = 0.675 | |
| **Comfortability (1–100)** | 80 (20) | 100 (0) | 80 (10) | 90 (20) | 80 (20) | <0.0001 |
| Comparison* | | p<0.001 | p = 0.017 | p<0.001 | p = 0.495 | |

* 2-hour measurement compared with 4 other measurements (Mann Withney U test)

** Comparisons between all measurements (Kruskal Wallis test)

## Discussion

In this study, we aimed to quantify the cognitive load experienced by residents working in the ED and its distribution during the day. During Delphi sessions, three scales were chosen to measure cognitive load: the complexity, the mental effort and the comfortability scale. To our surprise, the experienced cognitive load when assessed per decision was low. Because the cognitive load may be determined by an accumulation of decisions rather than an individual decision, it was decided to measure load over longer time periods (1-hour, 2-hour, per patient, per shift) as well. Indeed, the scores over these longer time periods were significantly higher than when measured per decision with the exception of the scores for the complexity scale, which remained low. The highest mental effort (score 1–9) and lowest comfortability scores (0–100%) were found for the measurements per shift (5.5 and 80%, resp.) and per 2-hour interval (4.0 and 80%, resp.). Using the 2-hour measurement as reference, the time periods between 21h-23h yielded the highest mental effort. Mental effort increased somewhat earlier, from 17h onwards, as comfortability decreased from 21h until the end of the evening shift. Median complexity (score 1–3) slightly increased during the day, until a score of 2 was reached at 19h.

We found a higher experienced cognitive load for the measurements that included a longer time period than those per decision or per patient. It can be argued that during a complete shift more factors contribute to the cognitive load than the individual decision. These factors may be interruptions, accumulation of decisions over a longer period, duration of the shift, being engaged with more than one patient at a time and the severity of cases [3, 4]. The measurements per decision and even per 1-hour interval cover a smaller burden with less influencing factors than per 2-hour interval. This finding contradicts the peak-end effect, which is a psychological phenomenon where people will grade an experience as more difficult when asked directly after the experience, compared to when asked later on [13].

The comfortability score in our study was overall high (median 80–100%). Although it is possible that the residents genuinely feel rather comfortable, it may be plausible that this score is influenced by the feeling that residents cannot admit to being uncomfortable since this may reflect failure or moral uncertainty [14]. This feeling may also affect the scores of the two other scales we used in our study.

The score for complexity was unexpectedly low and hardly varied during the day. The questions arise whether residents experience working in the ED as not complex, whether the complexity scale needs more categories than 3 to quantify complexity and whether this scale is able to quantify cognitive load. The scale is not widely used in the medical field.

The mental effort scale was designed as a measure of cognitive load [12] and in our opinion, illustrates the cognitive load better than the complexity and comfortability scale. Using this

scale, we found more variations in scores between the different types of measurements and more fluctuations during the day, which is to be expected. This may indicate that the mental effort scale has a better discriminatory ability than the other two scales. Mental effort was rated higher in a Canadian ED than in our study; median 5.5 for residents [15]. The patient load per physician in that study was, however, probably higher than in our ED. In our study, the mental effort scores were often exceeding 6 (33.3% of 2-hour and shift measurements; data not shown) as well, mostly during the busiest hours of the day. Mental effort was also scored 6 by paediatric residents in another recent study [16]. In both studies, mental effort was scored after longer periods, after a several hours or after the shift, respectively. Other studies recommend questionnaires (self-report measures) or measurements, such as eye tracking, to assess cognitive load [17, 18], but these studies focused on professional activities, such as surgical procedures, or concerned different professionals such as pilots and police officers.

The scores for the patient and shift measurements were slightly higher in two centres for complexity and mental effort, and lower for comfortability than in the third centre. This could be due to differences in experience of the residents, which was lower in the two centres with higher scores.

The experienced cognitive load varied over the day, with higher complexity and mental effort and lower comfortability in the evening (17h-23h) for the reference measurement (2-hour measurement). While the complexity and the comfortability score remained respectively low and high, the mental effort scores showed considerable variations in the afternoon and evening (17h-23h). The rise in mental effort scores started at 17h. Around this time, shifts change, patients are handed over, the number of residents decrease and direct supervision stops, while patient influx increases [4, 5]. Moreover, it is plausible that the caseload in the evening shifts is more complex, compared to daytime. This could lead to situations exceeding cognitive load capacity and thereby decreased work performance. A higher probability of making mistakes was found in another study at the moment of handoffs [5]. Possible measures to alleviate cognitive load include adding extra (experienced) residents or supervisors during peak hours or changing shift times in such a way that residents do not start or stop at highly demanding moments [4]. Other measures may focus on decreasing extraneous load by presenting information to the residents in a more accessible way (for instance pop ups when results are aberrant) or to reduce background noise in the ED [19], decreasing intrinsic load (match complexity of patients to experience level of the resident), or decreasing germane load (train the residents to automise schemas, like the ABCDE methodology, and to connect new information to what they already know) [15].

The mental effort scale appears to measure experienced cognitive load better than the complexity and comfortability scales, but further research is required to: [1] investigate whether other scales or techniques (including timing of assessing cognitive load) are more adequate for analysing cognitive load and [2] relate cognitive load to other factors such as patient load and triage code.

As far as we know, this is one of the first studies to investigate the cognitive load experienced by residents working in the ED [15, 16]. Measuring cognitive load in the ED is important, as there is an inverse relationship between cognitive load and performance and learning efficiency [9, 12]. The mental effort scores found in our study generally fell within the range (3–6) of optimal learning [9], but in the evening, there were periods during which the mental effort score was higher than 6.

Although our study revealed results in line with our hypothesis, we state that more research is needed to establish the best way to measure cognitive load in this setting, and its influence on accuracy of decision making, self-confidence, job-satisfaction, mental health of residents and patient outcomes. We observed residents in 3 centres to increase external validity, but

there were some differences in experience levels between the centres. In addition, the number of measurements was relatively small and did not cover all hours of the day, which may limit generalisability and hinders sub analysis, e.g. regarding the impact of experience of residents on the cognitive load. However, these time-consuming measurements were performed in real-time situations, providing a unique dataset with an actual representation of the daily practice of the ED during the busiest hours.

## Conclusion

Measuring cognitive load experienced by residents working in the ED is challenging, but working in an ED requires moderate to high mental effort with a comfortability score that ranged from 80–100%. Especially when measured per 2-hour interval and per shift, the cognitive load was higher than when measured per decision, hour and patient. An unexpected finding was the low score on the complexity scale. The experienced cognitive load fluctuated and increased at the change of shifts and during the evening shift. These results may offer tools to organise the ED in such a way that cognitive load is not exceeding the cognitive capacity of the residents.

## Supporting information

**S1 Dataset.**
(SAV)

## Author Contributions

**Conceptualization:** Yvonne Groenen, Dirk Notten, Anique De Bruin, Jacqueline Buijs, Harm R. Haak, Hella F. Broggreve, Lars Lambriks, Patricia M. Stassen.

**Data curation:** Yael Appelboom, Patricia M. Stassen.

**Formal analysis:** Yael Appelboom, Yvonne Groenen, Dirk Notten.

**Investigation:** Yael Appelboom, Yvonne Groenen, Dirk Notten, Lars Lambriks, Patricia M. Stassen.

**Methodology:** Yael Appelboom, Yvonne Groenen, Dirk Notten, Anique De Bruin, Jacqueline Buijs, Harm R. Haak, Hella F. Broggreve, Lars Lambriks, Patricia M. Stassen.

**Project administration:** Patricia M. Stassen.

**Supervision:** Anique De Bruin, Jacqueline Buijs, Harm R. Haak, Patricia M. Stassen.

**Visualization:** Lars Lambriks.

**Writing – original draft:** Yael Appelboom, Yvonne Groenen, Dirk Notten, Patricia M. Stassen.

**Writing – review & editing:** Yael Appelboom, Yvonne Groenen, Anique De Bruin, Jacqueline Buijs, Harm R. Haak, Hella F. Broggreve, Lars Lambriks, Patricia M. Stassen.

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
