## [Decision Letter · Decision Letter 0]

4 Sep 2024

PONE-D-23-36048Experienced cognitive load in the emergency departmentPLOS ONE

Dear Dr. Stassen,

Thank you for submitting your manuscript to PLOS ONE. After careful consideration, we feel that it has merit but does not fully meet PLOS ONE’s publication criteria as it currently stands. Therefore, we invite you to submit a revised version of the manuscript that addresses the points raised during the review process. Please submit your revised manuscript by Oct 19 2024 11:59PM. If you will need more time than this to complete your revisions, please reply to this message or contact the journal office at plosone@plos.org. Please include the following items when submitting your revised manuscript:A rebuttal letter that responds to each point raised by the academic editor and reviewer(s). You should upload this letter as a separate file labeled 'Response to Reviewers'.A marked-up copy of your manuscript that highlights changes made to the original version. You should upload this as a separate file labeled 'Revised Manuscript with Track Changes'.An unmarked version of your revised paper without tracked changes. You should upload this as a separate file labeled 'Manuscript'.

We look forward to receiving your revised manuscript.

Kind regards,

Claudia Bull

Academic Editor

PLOS ONE

Journal Requirements:

2. In the online submission form you indicate that your data is not available for proprietary reasons and have provided a contact point for accessing this data. Please note that your current contact point is a co-author on this manuscript. According to our Data Policy, the contact point must not be an author on the manuscript and must be an institutional contact, ideally not an individual. Please revise your data statement to a non-author institutional point of contact, such as a data access or ethics committee, and send this to us via return email. Please also include contact information for the third party organization, and please include the full citation of where the data can be found.

**Additional Editor Comments:**

Thank you for the opportunity to review your manuscript entitled "Experienced cognitive load in the emergency department" for consideration in PLOS One. The study sought to investigate the cognitive load experienced by residents working in the ED, and whether this load differed throughout the day. In recognition of the increased patient- and workloads EDs the world over face, I found this to be a well written manuscript that presents novel findings likely of interest to emergency staff.

Both reviewers have highlighted the need for greater clarity in your methods section, and have provided critical input to strengthen your discussion. If you choose to revise your article for consideration in PLOS One, please also upload the STROBE checklist for this study to illustrate that you have adhered to the reporting guidelines for observation research.

Reviewers' comments:

Reviewer's Responses to Questions

**Comments to the Author**

1. Is the manuscript technically sound, and do the data support the conclusions?

Reviewer #1: Partly

Reviewer #2: Yes

2. Has the statistical analysis been performed appropriately and rigorously? 

Reviewer #1: Yes

Reviewer #2: I Don't Know

3. Have the authors made all data underlying the findings in their manuscript fully available?

Reviewer #1: Yes

Reviewer #2: No

4. Is the manuscript presented in an intelligible fashion and written in standard English?

Reviewer #1: Yes

Reviewer #2: Yes

5. Review Comments to the Author

Reviewer #1: Dear Editor,

Thank you for the opportunity to review this manuscript. This study investigated the cognitive load experienced by residents working in the ED and the differences in cognitive load throughout the day. They found overall low mental effort, low complexity and high comfortability. The manuscript is well written and the topic is interesting and relevant. However, I have some concerns about the data and the conclusions drawn from it. I would suggest that the following aspects should be considered.

Abstract

- Methods: "3 three" scales appears to be a typographical error.

- Methods: Perhaps also provide what is low and what is high for mental effort and comfort to be consistent.

- Conclusion: Reading the abstract, I feel that the conclusion is not fully consistent with the findings reported in the results. Cognitive load is presented as high, but is (only) 4. Comfortability is presented as low, but is 80%. These figures don't seem alarming to me. It is only when you look at the shift that the mental effort is at the higher end of the scale (but still only 5.5). I think it is more correct to mention the observed increase in mental effort in the evenings, as this is also something that can be used to organise shifts.

- However, a general question remains for me: What is high cognitive load? When is a person's cognitive load exceeded, and how is that defined? I think this needs to be put into perspective to fully understand these measurements (in the main text of the manuscript). Also, I think that working in the ED inherently leads to high(er) cognitive load as it is a demanding job - how does this resonate with other jobs/professions? Perhaps the results are actually quite normal and not worrying? Could this be reflected on more in the discussion section?

- It is also stated that the residents felt uncomfortable when making decisions, and based on this I would expect lower comfortability scores. Again, this needs to be put into perspective.

Introduction

- The introduction is well written and clearly describes the relevance and purpose.

- For me personally, the extraneous cognitive load and the germane load could use a few more words of explanation (in the way that the intrinsic load was described). Perhaps give an example based on an ED shift: when working in the ED, what is the germane load? I may have missed this or misunderstood, but does this come up elsewhere in the manuscript? Maybe this is something to reflect on in the discussion? Or maybe it could be left out as it just confused me.

- Why did the study only focus on residents (in internal medicine)?

Methods

- I think it would be good to have some more clarity on how the grading process went for the residents. When did they assess the cognitive load, was it immediately after a decision? And how did they do it? Were they asked by the researchers? Did they have to fill in a form? Was every decision reported, or could some have been missed?

- It would be interesting to include the scales in some supplementary material, as not everyone (especially the clinically oriented audience) will be familiar with these scales, and not all of them were easily found online.

- Were the scales available in Dutch or were they translated by the authors? How did they do this?

- Consider giving the total number of weeks that the observations took place.

- What is your rationale for looking for differences between the five types of measurement compared to the 2-hour measurement (see also my comment in the Results section).

- Explanation: Why did you look at the correlation between the three scales? What does this tell us?

Results

- Table 1: Why are all the percentages in the most right column 20%?

- I have some concerns about the generalisability of the results, especially given the small number of residents in teaching hospitals and the lack of day shifts and few evening shifts (on which most of the conclusions are based). When I read Table 4, I see that the overall complexity is low, the mental effort low to medium, and the comfortability high. Should the conclusions about high cognitive load be a little more cautious, given the number of participants and the fact that not all shifts were observed?

- Complementary measurements: the comfortability score was lower after 15h, how much lower? Is this a clinically relevant difference?

- You mention no differences in the complementary measurements (time interval measurements) and then in the next section (comparison of measurements) there are differences. I do not fully understand Table 5 and how this contributes. Why do you compare the medians with the 2-hour interval as a reference? Why did you choose to compare the different measurements to the 2 hour reference? Is this the best way to say something about high cognitive load? Wouldn't you expect everything to be higher/worse for a 2 hour period combined compared to just individual decisions/patients? Maybe I misunderstood, but in that case I think it needs more explanation.

- Did you see a difference per hospital or per year of experience? Or according to the type of decision (e.g. evaluation of results might be more demanding than delegation or patient registration). Could these factors explain some of the differences?

- A small typo in Table 5 (measurements)

Discussion

- You conclude: comfortability was overall high (80-100%) – this is not consistent with the abstract.

- I have some concerns about the generalisability of the results, given the number of participants and the number of shifts. I think this should be reflected in the discussion section.

- When is cognitive load exceeded? How is it defined? These results are quite reassuring when I read the manuscript and I would like to see some more elaboration on this (see also my comment on this in the abstract section).

- I would find it interesting if the discussion were a bit more elaborate on how exactly these results would influence practice. Would you recommend changing the way shifts are organised now? What other recommendations might there be? It would be interesting to read some thoughts on this.

- Explain a bit more about other studies that have been done with these scales and how well they measure, for example, complexity.

- I think it is a strength that a Delphi meeting took place.

- I appreciate the effort to do these measurements in a busy ED in real time situations, which provides a unique data set. This is a strength of the study.

All in all, this study was interesting to read and provided insights into the cognitive load of residents in internal medicine working in the emergency department. However, both the methods and some of the conclusions need to be rationalised to make them more understandable and accurate. There are some other results that might be interesting to look at (differences by years of experience, by type of decision, etc.).

Reviewer #2: This is an interesting prospective, multicenter study involving three hospitals, focused on the cognitive load experienced by residents in the emergency department and how it varies throughout the day. While the results section is clear, the study's procedures are not explained in sufficient detail, particularly regarding practical aspects.

More information is needed in the "Methods" section:

I don’t fully understand how the Emergency Departments are organized in these three hospitals and what the residents' responsibilities are. Do they only handle outpatients, or do they also manage short-term hospitalizations (before traditional hospitalization or discharge)? Do they work in the resuscitation room? How many residents are there per shift, and how many patients do they manage? Please describe the workload and any differences between the three hospitals.

• What kind of supervision was provided (or was there no supervision)? Was the supervision consistent across the three hospitals?

• How were the data collected by the investigators? Did the residents fill out questionnaires, have conversations with the investigators, or were they observed?

• Regarding "Decision":

o Please explain why "patient registration" is considered a decision.

o Clarify the "Delegation of tasks": Is it from the resident to someone else (to whom?) or from someone else to the resident?

• How were the shifts selected for the study? Was it one shift per day, or were they selected randomly?

• Regarding the weekly meetings: Who attended these meetings (residents, investigators from the 3 centers)?

• How many investigators participated in the entire study?

• Why did you choose to measure cognitive load using not only mental effort but also complexity and comfort?

Results:

• Page 7, "Complementary results": There is a repetition at the end of the first paragraph.

Discussion:

• Page 9, 2nd paragraph:

o Could you comment based on cognitive load theory? Specifically, what about the extraneous load?

o The "peak-end effect" is difficult to understand clearly because it’s not clear how and when the data were collected.

• Page 9, 3rd paragraph:

o "Residents cannot admit to being uncomfortable": You could have discussed this point by comparing the data of more experienced residents with that of less experienced residents, as you mention later in the discussion. This comparison data would be interesting to include in the "Results" section.

Example of another study on cognitive load in the Emergency Room:

• Impact of training in the supervision of clinical reasoning in the pediatric emergency department on residents' perception of the on-call experience. Pietrement C, Barbe C, Bouazzi L, Maisonneuve H. Arch Pediatr. 2023 Nov;30(8):550-557. doi: 10.1016/j.arcped.2023.08.006.

6. PLOS authors have the option to publish the peer review history of their article (what does this mean?). If published, this will include your full peer review and any attached files.

Reviewer #1: No

Reviewer #2: No

---

## [Author Response · Author response to Decision Letter 0]

14 Oct 2024

PONE-D-23-36048

Experienced cognitive load in the emergency department

Dear editor and dear reviewers,

Hereby, we resubmit a revised manuscript “Experienced cognitive load in the emergency department” (PONE-D-23-36048). We have read your thorough review and adjusted our mansucript accordingly. We think that the review process has led to significant improvements of the paper, and we thank you for your remarks.

Please see below for our responses to your remarks and for the changes we made.

Additional Editor Comments:

Thank you for the opportunity to review your manuscript entitled "Experienced cognitive load in the emergency department" for consideration in PLOS One. The study sought to investigate the cognitive load experienced by residents working in the ED, and whether this load differed throughout the day. In recognition of the increased patient- and workloads EDs the world over face, I found this to be a well written manuscript that presents novel findings likely of interest to emergency staff.

Both reviewers have highlighted the need for greater clarity in your methods section, and have provided critical input to strengthen your discussion. If you choose to revise your article for consideration in PLOS One, please also upload the STROBE checklist for this study to illustrate that you have adhered to the reporting guidelines for observation research.

Thank you. Please find our responses below, and in the revised manuscript.

Reviewers' comments:

Reviewer's Responses to Questions

Comments to the Author

1. Is the manuscript technically sound, and do the data support the conclusions?

Reviewer #1: Partly

Reviewer #2: Yes

2. Has the statistical analysis been performed appropriately and rigorously? 

Reviewer #1: Yes

Reviewer #2: I Don't Know

3. Have the authors made all data underlying the findings in their manuscript fully available?

The requires authors to make all data underlying the findings described in their manuscript fully available without restriction, with rare exception (please refer to the Data Availability Statement in the manuscript PDF file). The data should be provided as part of the manuscript or its supporting information, or deposited to a public repository. For example, in addition to summary statistics, the data points behind means, medians and variance measures should be available. If there are restrictions on publicly sharing data—e.g. participant privacy or use of data from a third party—those must be specified.

Reviewer #1: Yes

Reviewer #2: No

4. Is the manuscript presented in an intelligible fashion and written in standard English?

Reviewer #1: Yes

Reviewer #2: Yes

5. Review Comments to the Author

Reviewer #1: Dear Editor,

Thank you for the opportunity to review this manuscript. This study investigated the cognitive load experienced by residents working in the ED and the differences in cognitive load throughout the day. They found overall low mental effort, low complexity and high comfortability. The manuscript is well written and the topic is interesting and relevant. However, I have some concerns about the data and the conclusions drawn from it. I would suggest that the following aspects should be considered.

Abstract

- Methods: "3 three" scales appears to be a typographical error. 

You are correct; we adjusted the text.

- Methods: Perhaps also provide what is low and what is high for mental effort and comfort to be consistent.

We adjusted the text accordingly.

- Conclusion: Reading the abstract, I feel that the conclusion is not fully consistent with the findings reported in the results. Cognitive load is presented as high, but is (only) 4. Comfortability is presented as low, but is 80%. These figures don't seem alarming to me. It is only when you look at the shift that the mental effort is at the higher end of the scale (but still only 5.5). I think it is more correct to mention the observed increase in mental effort in the evenings, as this is also something that can be used to organise shifts.

- However, a general question remains for me: What is high cognitive load? When is a person's cognitive load exceeded, and how is that defined? I think this needs to be put into perspective to fully understand these measurements (in the main text of the manuscript). Also, I think that working in the ED inherently leads to high(er) cognitive load as it is a demanding job - how does this resonate with other jobs/professions? Perhaps the results are actually quite normal and not worrying? Could this be reflected on more in the discussion section?

When cognitive load is too high for an individual, the performance will decrease. A score on the mental efforts scale within 3-6 is within the range of optimal performance/learning. So, we agree on your remark on the word “high”. We replaced this word by “higher”. We added a reflection on the meaning of the number 4 to the discussion.

We adjusted the text in the abstract to:

“In contrast, the cognitive load was higher per 2-hour interval (mental effort: median 4.0 (IQR 4.0) and comfortability 80% (IQR 20), resp.) and per shift (mental effort: median 5.5 (IQR 4.0) and comfortability 80% (IQR 20).”

and to:

“From 17h onwards, residents working in the ED experienced rather high mental effort and reported feeling not optimally uncomfortable when making decisions”

We added some reflections on this issue in the discussion.

“Measuring cognitive load in the ED is important, as there is an inverse relationship between cognitive load and performance and learning efficiency (9, 12). The mental effort scores found in our study generally fell within the range (3-6) of optimal learning (9), but in the evening, there were periods during which the mental effort score was higher than 6.”

- It is also stated that the residents felt uncomfortable when making decisions, and based on this I would expect lower comfortability scores. Again, this needs to be put into perspective.

Agreed. See the adjustments made above.

Introduction

- The introduction is well written and clearly describes the relevance and purpose.

- For me personally, the extraneous cognitive load and the germane load could use a few more words of explanation (in the way that the intrinsic load was described). Perhaps give an example based on an ED shift: when working in the ED, what is the germane load? I may have missed this or misunderstood, but does this come up elsewhere in the manuscript? Maybe this is something to reflect on in the discussion? Or maybe it could be left out as it just confused me.

We added a short sentence to explain germane load. And added some examples to the Discussion.

“The intrinsic cognitive load is determined by the complexity of a task and the person’s knowledge. The extraneous cognitive load is created by the way a task is presented, and the germane load is the result of the construction and automation of schemes, eg. patterns that can be recognized”. 

“Other measures may focus on decreasing extraneous load by presenting information to the residents in a more accessible way (for instance pop ups when results are aberrant) or to reduce background noise in the ED (19), decreasing intrinsic load (match complexity of patients to experience level of the resident), or decreasing germane load (train the residents to automise schemas, like the ABCDE methodology, and to connect new information to what they already know) (15).”

- Why did the study only focus on residents (in internal medicine)?

We focused on one specialty because this project is part of an evaluation of the educational activities that we organize for our residents. Residients from other specialites may experience other cognitive load as their work (load) differs.

Methods

- I think it would be good to have some more clarity on how the grading process went for the residents. When did they assess the cognitive load, was it immediately after a decision? And how did they do it? Were they asked by the researchers? Did they have to fill in a form? Was every decision reported, or could some have been missed?

This is indeed important to add to the methods. 

All residents were asked by the researchers immediately after the decision, hour, shift etc.

We added the following sentence to the methods section.

“The residents were asked regarding the three scales by the investigators immediately after the decision was made, or after the different time-intervals.”

- It would be interesting to include the scales in some supplementary material, as not everyone (especially the clinically oriented audience) will be familiar with these scales, and not all of them were easily found online.

The three scores are all Likert scales, without other rubrics than mentioned in the text:

The complexity scale (score 1-3), where 1 represents a simple and 3 a highly complicated problem (8). 

The second scale was the mental effort scale (score 1-9), where 1 is scored for the lowest, and 9 for the highest mental effort (11). 

The third scale was the comfortability scale, which is a percentage scale, with 100% being completely comfortable and 0% being not at all comfortable (12).

- Were the scales available in Dutch or were they translated by the authors? How did they do this? 

The residents were interviewed in Dutch, as mentioned above, without more rubrics than mentioned in the text. Because of the simplicity of the scores it was easy to translate the scores.

- Consider giving the total number of weeks that the observations took place.

We observed 45 shifts between September and December 2019, as stated in the methods section.

- What is your rationale for looking for differences between the five types of measurement compared to the 2-hour measurement (see also my comment in the Results section).

This is due to advancing insights. We started rating decisions, and found a) a very low cognitive load, and b) no variation over time, which was not to be expected. That’s why we introduced more measurements after Delphi discussions.

- Explanation: Why did you look at the correlation between the three scales? What does this tell us?

To date, there is no gold standard to measure cognitive load, we chose 3 scores to measure different aspects of cognitive load and because it is possible that one score measures cognitive load better than the other.

We found it interesting to see how the differents scores were interconnected. That is the reason why we calculated the correlations, like others did (eg Vella, ref 15).

Results

- Table 1: Why are all the percentages in the most right column 20%?

That is because there is a typo. Thank you for noting this. The first 20% should have been 60%. 

After looking at table 1, we discovered that Table 1 and 2 were not alligned, because MMC should have been Zuyderland in table 1. We corrected Table 2. 

- I have some concerns about the generalisability of the results, especially given the small number of residents in teaching hospitals and the lack of day shifts and few evening shifts (on which most of the conclusions are based). 

We scored the answers of 14 residents during 45 shifts in 3 centres. Around 40% of observed shifts involved the most busy moment of the day, after 17h.

This shadowing of the residents was very labor-intensive for the researchers. As stated in our discussion, we mentioned this limitation, and we agree on this.

“In addition, the number of measurements was relatively small and did not cover all hours of the day, which may limit generalisability and hinders subanalyses, eg. regarding the impact of experience of residents on the cognitive load.”

When I read Table 4, I see that the overall complexity is low, the mental effort low to medium, and the comfortability high. Should the conclusions about high cognitive load be a little more cautious, given the number of participants and the fact that not all shifts were observed?

See adjustments made in response to your earlier remark on this topic.

- Complementary measurements: the comfortability score was lower after 15h, how much lower? Is this a clinically relevant difference? 

We added the median values to this section (80 vs 90%).

- You mention no differences in the complementary measurements (time interval measurements) and then in the next section (comparison of measurements) there are differences. I do not fully understand Table 5 and how this contributes. Why do you compare the medians with the 2-hour interval as a reference? Why did you choose to compare the different measurements to the 2 hour reference? Is this the best way to say something about high cognitive load? Wouldn't you expect everything to be higher/worse for a 2 hour period combined compared to just individual decisions/patients? Maybe I misunderstood, but in that case I think it needs more explanation.

This analysis is indeed somewhat complex. 

The first section, time interval measurements, aims to show variation over the day for 1-h and 2-h measurements. 

The second section is on the comparison between the five ways we scored cognitive load (1h, 2h, decision, patient, shift). We think this analysis adds because you can see that the median cognitive load varies between the measurements. So if you measure cognitive load per decision, the load is very low, while the load is high when you measure per shift.

We decided to select the 2h measurement as a reference as we retrieved relatively many data on this measurement in all three centres. 

- Did you see a difference per hospital or per year of experience? Or according to the type of decision (e.g. evaluation of results might be more demanding than delegation or patient registration). Could these factors explain some of the differences?

Although these topics are interesting, our aim was to focus on measuring cognitive load in a prospective, real-time way. All analyses per decision showed very low cognitive load (4.6% mental effort>6). For experience subanalyses, our data did not have enough power.

This is added to the discussion (limitations).

- A small typo in Table 5 (measurements)

We are not sure which typo you have found.

Discussion

- You conclude: comfortability was overall high (80-100%) – this is not consistent with the abstract. 

We adjusted the abstract.

- I have some concerns about the generalisability of the results, given the number of participants and the number of shifts. I think this should be reflected in the discussion section.

We now made this limitation more clear.

“In addition, the number of measurements was relatively small and did not cover all hours of the day, which may limit generalisability.”

- When is cognitive load exceeded? How is it defined? These results are quite reassuring when I read the manuscript and I would like to see some more elaboration on this (see also my comment on this in the abstract section).

We agree on your remark. We added a cut off of 6 for mental effort where learning/performance will not be optimal. We added this to the discussion as it was not our aim at the start of the study.

“Measuring cognitive load in the ED is important, as there is an inverse relationship between cognitive load and performance and learning efficiency (9, 12). The mental e

---

## [Decision Letter · Decision Letter 1]

1 Nov 2024

PONE-D-23-36048R1Experienced cognitive load in the emergency department. A prospective study.PLOS ONE

Dear Dr. Stassen,

Thank you for submitting your manuscript to PLOS ONE. After careful consideration, we feel that it has merit but does not fully meet PLOS ONE’s publication criteria as it currently stands. Therefore, we invite you to submit a revised version of the manuscript that addresses the points raised during the review process. **The reviewers only have additional minor comments for you to address. **

We look forward to receiving your revised manuscript.

Kind regards,

Claudia Bull

Academic Editor

PLOS ONE

**Journal Requirements:**

Reviewers' comments:

Reviewer's Responses to Questions

**Comments to the Author**

1. If the authors have adequately addressed your comments raised in a previous round of review and you feel that this manuscript is now acceptable for publication, you may indicate that here to bypass the “Comments to the Author” section, enter your conflict of interest statement in the “Confidential to Editor” section, and submit your "Accept" recommendation.

Reviewer #1: All comments have been addressed

Reviewer #2: (No Response)

2. Is the manuscript technically sound, and do the data support the conclusions?

Reviewer #1: Yes

Reviewer #2: Yes

3. Has the statistical analysis been performed appropriately and rigorously? 

Reviewer #1: Yes

Reviewer #2: I Don't Know

4. Have the authors made all data underlying the findings in their manuscript fully available?

Reviewer #1: Yes

Reviewer #2: Yes

5. Is the manuscript presented in an intelligible fashion and written in standard English?

Reviewer #1: Yes

Reviewer #2: Yes

6. Review Comments to the Author

**Reviewer #1:** I think the authors have improved their manuscript by using the reviewer comments. The findings have been put into more perspective and the conclusions are more appropriate. I also think it is suitable for acceptance after some very minor updates.

Some minor remarks: (1) The sentence that was added on page 5 about the data collection: "The residents were asked regarding the cognitive load by the investigators immediately after the decision was made, or immediately after the different time-intervals." Does that mean that sometimes the residents were asked after a certain time interval (e.g. 2 hours for example) and had to recall all decisions they made? That is something different tan being asked immediatly after a decision. This might need some extra clarification. (2) Under table 5 there is the typo 'measurments'. (3) The complementary measurements you admit are somewhat complex, you provide an explanation to me as a reviewer, but you might want to add it to the methods section for readers to make it clearer.

**Reviewer #2:** I thank the authors for the revised manuscript. The comments have been addressed.

I just have few new comments :

- Setting and study design : In MUMC, the supervisor is present until 8 pm, while in MMC, emergency physicians assist the residents if necessary. What about Zuyderland hospital ?

Results :

- The median complexity for all five types of measurements was comparable (1.0), although differences in the distribution of the scores resulted in significant differences : I suggest to add a supplement material to see the distribution

- mental effort scores were often exceeding 6 (33.3% of 2-hour and shift measurements; data not shown) : I suggest to add a supplement material to see the percentages

7. PLOS authors have the option to publish the peer review history of their article (what does this mean?). If published, this will include your full peer review and any attached files.

Reviewer #1: No

Reviewer #2: No

---

## [Author Response · Author response to Decision Letter 1]

2 Nov 2024

Dear editor and reviewers,

please see our response below.

Thank you for your review. See our atteched manuscript for lay-out figures

Reviewer #1: I think the authors have improved their manuscript by using the reviewer comments. The findings have been put into more perspective and the conclusions are more appropriate. I also think it is suitable for acceptance after some very minor updates.

Some minor remarks: 

(1) The sentence that was added on page 5 about the data collection: "The residents were asked regarding the cognitive load by the investigators immediately after the decision was made, or immediately after the different time-intervals." Does that mean that sometimes the residents were asked after a certain time interval (e.g. 2 hours for example) and had to recall all decisions they made? That is something different tan being asked immediately after a decision. This might need some extra clarification.

Thank you for your second review and compliments. 

Responding to your first remark:

“The residents were asked regarding the three scales by the investigators immediately after the decision was made, or after the different time-intervals.” was stated in the methods. 

Depending on the type of measurement, it is logical to ask the questions either after each decision, or after each shift, 1-2hour interval, etc..

We therefore added: …”the timing depending on the type of measurement”.

(2) Under table 5 there is the typo 'measurments'.

Thanks for noticing this. Text adjusted.

(3) The complementary measurements you admit are somewhat complex, you provide an explanation to me as a reviewer, but you might want to add it to the methods section for readers to make it clearer.

We added a sentence to the analysis section and to the text and Table subscript in the results to make this more clear to all readers.

Reviewer #2: I thank the authors for the revised manuscript. The comments have been addressed.

I just have few new comments :

- Setting and study design : In MUMC, the supervisor is present until 8 pm, while in MMC, emergency physicians assist the residents if necessary. What about Zuyderland hospital ?

The situation it the same. Text is adjusted in the methods.

Thank you for your second review.

Results :

- The median complexity for all five types of measurements was comparable (1.0), although differences in the distribution of the scores resulted in significant differences : I suggest to add a supplement material to see the distribution

We reanalysed the median complexity for the 5 measurements and came to the same conclusion. Kruskal-Wallis test, P value < 0.0001

We exported the analysis to GraphPad and created a new figure.

 1-h complex 2h-complex shift patient decision

Number of values 74 45 24 79 592

25% Percentile 1,000 1,000 1,000 1,000 1,000

Median 1,000 1,000 1,000 1,000 1,000

75% Percentile 2,000 2,000 2,000 2,000 1,000

The line at the bottom of the figure is the median, with IQR indicated with the lines.

We, however, think the figure does not add. If the reviewer wishes so we can add the figure to the supplementary information.

- mental effort scores were often exceeding 6 (33.3% of 2-hour and shift measurements; data not shown) : I suggest to add a supplement material to see the percentages

We made a new figure using GraphPad.

We don’t think adding the figure adds to the text, especially as this analysis concerns a subanalysis, which is mentioned in the discussion only. We did not prepare for this analysis in advance. If the editor wishes so, we are of course willing to add the material.

---

## [Editor Report · Decision Letter 2]

5 Nov 2024

Experienced cognitive load in the emergency department. A prospective study.

PONE-D-23-36048R2

Dear Dr. Stassen,

We’re pleased to inform you that your manuscript has been judged scientifically suitable for publication and will be formally accepted for publication once it meets all outstanding technical requirements.

Kind regards,

Claudia Bull

Academic Editor

PLOS ONE
---

## [Editor Report · Acceptance letter]

22 Nov 2024

PONE-D-23-36048R2 

PLOS ONE

Dear Dr. Stassen, 

I'm pleased to inform you that your manuscript has been deemed suitable for publication in PLOS ONE. Congratulations! Your manuscript is now being handed over to our production team.

Kind regards, 

on behalf of

Dr. Claudia Bull 

%CORR_ED_EDITOR_ROLE%

PLOS ONE